# Antichiral surface states in time-reversal-invariant photonic semimetals

Jian-Wei Liu[1], Fu-Long Shi[1], Ke Shen[1], Xiao-Dong Chen[1], Ke Chen[1], Wen-Jie Chen [1]✉ & Jian-Wen Dong [1]✉

Besides chiral edge states, the hallmark of quantum Hall insulators, antichiral edge states can exhibit unidirectional transport behavior but in topological semimetals. Although such edge states provide more flexibility for molding the flow of light, their realization usually suffers from time-reversal breaking. In this study, we propose the realization of antichiral surface states in a time-reversal-invariant manner and demonstrate our idea with a three-dimensional (3D) photonic metacrystal. Our system is a photonic semimetal possessing two asymmetrically dispersed Dirac nodal lines. Via dimension reduction, the nodal lines are rendered a pair of offset Dirac points. By introducing synthetic gauge flux, each two-dimensional (2D) subsystem with nonzero $k_z$ is analogous to a modified Haldane model, yielding a $k_z$-dependent antichiral surface transport. Through microwave experiments, the bulk dispersion with asymmetric nodal lines and associated twisted ribbon surface states are demonstrated in our 3D time-reversal-invariant system. Although our idea is demonstrated in a photonic system, we propose a general approach to realize antichiral edge states in time-reversal-invariant systems. This approach can be easily extended to systems beyond photonics and may pave the way for further applications of antichiral transport.

Chiral edge state, circulating around a gapped material in either clockwise or counterclockwise manner, is a fundamental feature of 2D quantum Hall systems[1–5]. In the past few decades, these unidirectional edge states have led to a revolutionized chapter in condensed matter physics and photonics[6–11], because of their promising applications in one-way waveguides[12–21], topological lasers[22–27], topological photonic circuits[28–31], etc. Besides gapped materials, unidirectional edge transport can be induced in gapless materials. One approach is to introduce a pseudomagnetic field by shifting the Dirac cones in momentum space with an inhomogeneous strained lattice[32–34]. Another approach is to shift the Dirac cones in frequency by considering a staggered magnetic flux in Haldane model[35–42]. In this type of 2D Dirac semimetal, two Dirac frequencies at K and K′ points shift in opposite directions, yielding a tilted antichiral edge dispersion. Unlike chiral edge states that counter-propagate along the parallel edges of a strip sample, these antichiral edge states copropagate along the parallel edges,

providing more flexibility for molding the flow of light. However, because frequency splitting between Dirac points naturally requires time-reversal breaking, its photonic realization is rather challenging due to intrinsic material loss and weak magneto-optic effect. It would be highly desirable if we could achieve antichiral edge transport without time-reversal breaking.

Interestingly, 2D Dirac semimetals are related to various higher-dimensional topological semimetals, such as 3D Dirac nodal line semimetals and their variants (nodal rings[43–47], nodal links[48,49], nodal chains[50–53]). Nodal line semimetals can be modeled as layers of 2D Dirac semimetals coupled in the stacking direction[54,55]. With the interlayer coupling, the two Dirac points disperse in the $k_z$ direction, forming two straight nodal lines[55–57]. This inspired us to consider an unexplored route to achieve antichiral edge transport. By engineering appropriate interlayer couplings, two Dirac nodal lines can evolve asymmetrically in frequency, yielding two offset Dirac points for a fixed nonzero $k_z$,

[1]School of Physics & State Key Laboratory of Optoelectronic Materials and Technologies, Sun Yat-sen University, Guangzhou 510275, China.
✉e-mail: chenwenj5@mail.sysu.edu.cn; dongjwen@mail.sysu.edu.cn

along with tilted edge states in between. Notably, the frequency offset between the two Dirac points does not embody the violation of time reversal symmetry because $\bar{K}$ and $\bar{K}'$ do not form a time-reversal pair for nonzero $k_z$.

In this letter, we propose a 3D layer-stacked photonic metacrystal possessing antichiral surface states, which has never been realized in time-reversal-invariant systems. Its interlayer couplings mimic an effective staggered magnetic flux. For any nonzero $k_z$, the 3D system reduces to a 2D modified Haldane model (MHM), with two Dirac cones split in frequency. From the perspective of 3D band dispersion, these correspond to two asymmetric nodal lines (ANLs) in 3D momentum space, along with a twisted ribbon surface state. Based on this idea, we design and fabricate a layer-stacked photonic metacrystal. The bulk dispersions of ANLs and associated antichiral surface states are experimentally demonstrated by the Fourier spectra of measured electric fields. The antichiral surface transport is confirmed by the measured surface transmission spectra.

## Results

### Tight-binding analysis

An modified Haldane model, where the magnetic fluxes acting on two sublattices take opposite signs (Fig. 1a), was originally proposed to exhibit the antichiral edge mode[35]. A major challenge to implementing such a model in a photonic system is the realization of a staggered magnetic flux, or equivalently, a nonreciprocal next-nearest-neighbor (NNN) coupling. Recent studies have reported that this issue can be overcome via gyromagnetic photonic crystals with precise on-site modulation of magnetization[38,40]. However, owing to the intrinsic material loss and weak magneto-optic effect of gyromagnetic materials, this scheme can hardly be extended to the optical regime.

To illustrate how our idea works, we start with a tight-binding model for an AA-stacked honeycomb lattice (Fig. 1b). Its photonic realization will be discussed later. The Hamiltonian of this 3D lattice can be expressed as follows:

$$H = \sum_{<i,j>;m} (t_1 a_{i,m}^\dagger b_{j,m} + h.c.) + \sum_{<<i,j>>;m} (t_2 a_{i,m}^\dagger a_{j,m+1} + t_2 b_{i,m}^\dagger b_{j,m+1} + h.c.),$$

(1)

where $a$ ($b$) and $a^\dagger$ ($b^\dagger$) are the annihilation and creation operators on sublattice sites, respectively, $i$ and $j$ label the position of the lattice in each layer, and $m$ is the layer index. For simplicity, the on-site energy difference between sublattices is neglected. Only the intralayer coupling $t_1$ [white sticks in Fig. 1b] and interlayer coupling $t_2$ [yellow sticks in Fig. 1b] are considered and are both real numbers as restricted by time reversal. After the Fourier transformation, the corresponding Bloch Hamiltonian has the following form:

$$H(\mathbf{k}) = \begin{pmatrix} \lambda_{\mathbf{k}} & t_1 \beta_{\mathbf{k}} \\ t_1 \beta_{\mathbf{k}}^* & \lambda_{\mathbf{k}} \end{pmatrix},$$

(2)

where $\beta_{\mathbf{k}} = 1 + \exp(ik_x a) + \exp(ik_x a/2 + i\sqrt{3}k_y a/2)$, and $a$ is the lattice constant;

$$\lambda_{\mathbf{k}} = t_2 \cos k_z d[\cos k_x a + \cos(k_x a/2 + \sqrt{3}k_y a/2) + \cos(k_x a/2 - \sqrt{3}k_y a/2)]$$
$$+ t_2 \sin k_z d[\sin k_x a - \sin(k_x a/2 + \sqrt{3}k_y a/2) - \sin(k_x a/2 - \sqrt{3}k_y a/2)]$$

(3)

represents the energy modulation by interlayer couplings and $k_z$, and $d$ is the interlayer distance. The first Brillouin zone (BZ) of the 3D lattice is shown in Fig. 1c.

Interestingly, because the proposed system is periodic in the $z$ direction, $k_z$ is a good quantum number. If we consider only the dispersion and transport in the $x$-$y$ plane, the 3D system can be reduced to an effective 2D system, and $k_z$ is regarded as an external parameter. Its

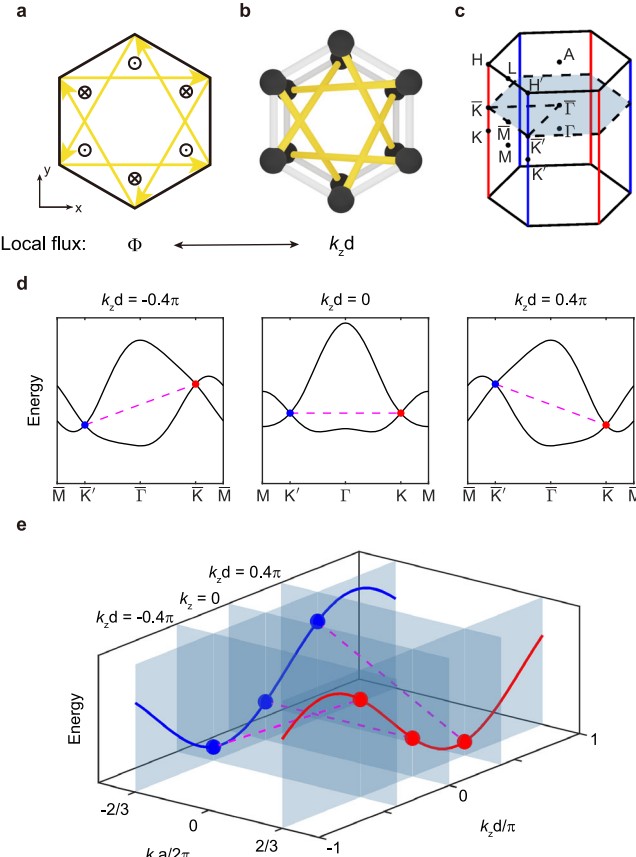

**Fig. 1 | Synthetic gauge flux in time-reversal system and the resulting energy-offset between Dirac points. a** 2D tight-binding model for a modified Haldane lattice. The NNN couplings are represented by yellow arrows, resulting in staggered magnetic flux $\Phi$ ("$\otimes$" and "$\odot$"). **b** 3D tight-binding model for an AA-stacked honeycomb lattice. Via dimension reduction, the interlayer couplings (yellow sticks) mimic nonreciprocal NNN couplings as in an MHM. In the reduced 2D system with a fixed $k_z$, the effective staggered magnetic flux is proportional to $k_z d$. **c** The first Brillouin zone of the 3D hexagonal lattice. A 2D cut plane with a specified $k_z$ is highlighted by a cyan plane, which corresponds to the 2D BZ of the reduced system. **d** Bulk band structures on different $k_z$ planes. The 2D subsystem hosts two iso-energy Dirac points for the zero $k_z$, whereas it hosts energy-offset Dirac points for nonzero $k_z$. **e** ANLs on $-HH$ and $-H'H'$ by solving the 3D tight-binding Hamiltonian.

2D BZ corresponds to a $k_z$ cut plane in the original 3D BZ (the cyan plane in Fig. 1c). The original interlayer coupling $t_2$ is reduced to an in-plane NNN coupling with a complex coefficient of $t' = t_2 \exp(\pm ik_z d)$, where the sign of the argument depends on the direction of coupling. Then, the system becomes an MHM; the yellow arrows in Fig. 1a indicate the direction along which the phase in $t'$ takes a positive sign. Via this dimension reduction approach, any type of local magnetic flux with a complex configuration can be implemented in a 3D time-reversal-invariant system.

Because the Pierels phase factor in $t'$ is proportional to $k_z$, the 3D system mimics different MHMs for different values of $k_z$. When $k_z = 0$, the bulk band structure (middle panel in Fig. 1d) exhibits two Dirac points with identical energy of $-3t_2/2$, as the synthetic gauge flux vanishes. In addition, a flat edge band (magenta dashed line) is expected to span their projections onto the surface BZ. This agrees with the fact that the $k_z = 0$ plane is a time-reversal-invariant $k$-plane in the 3D BZ. However, for a nonzero $k_z$ (left and right panels in Fig. 1d), the two Dirac points split in energy as the synthetic gauge flux turns on. Then, a tilted edge band pinned at the two Dirac points is expected. Notably, time reversal symmetry is preserved in our 3D system, meaning that the 2D subsystem with positive $k_z$ is the time-reversal

counterpart of the one with negative $k_z$. Thus, the energy splitting for $k_z = 0.4\pi/d$, as well as the propagation direction of antichiral edge states, is opposite to the case for $k_z = -0.4\pi/d$ (Fig. 1d).

From the perspective of 3D band structure, a series of Dirac points lying at different $k_z$ planes will form two straight nodal lines pinned to the edges of the hexagonal BZ (the red and blue lines in Fig. 1c). Owing to the lack of mirror symmetry in the $x$ direction, the two nodal lines have opposite energy dispersions along $k_z$. The eigenenergies of the two nodal lines are expressed as

$$E(\bar{K}/\bar{K}') = -3t_2 \cos(k_z d)/2 \mp 3\sqrt{3}t_2 \sin(k_z d)/2. \quad (4)$$

Meanwhile, the edge bands for different $k_z$ values (the dashed lines in Fig. 1e) together form a surface band with a ribbon shape, covering the rectangular region between two ANLs. As the nodal lines disperse asymmetrically in $k_z$ direction, the ribbon is tilted to the left in positive $k_z$ regime while tilted to the right in negative $k_z$ regime, yielding a "twisted ribbon" shape. Hence, the proposed 3D lattice model is a nodal line semimetal and the antichiral edge/surface states are the direct consequences of the asymmetric dispersions of the two Dirac nodal lines.

## Photonic metacrystals supporting antichiral surface states and asymmetric nodal lines

We now consider the real structure of photonic semimetals with two ANLs. Because electromagnetic waves can propagate, rather than decay, in dielectric media, the tight-binding model is usually not a good approximation for a real photonic structure. The aforementioned tight-binding lattice cannot be used as a design tool, but it can still guide us to photonic structures possessing the correct symmetry to exhibit ANLs. The symmetry of the studied tight-binding model belongs to a 3D hexagonal lattice with space group $P\bar{3}1m$ (no. 162), which has inversion symmetry and in-plane $C_3$ rotation symmetry. As illustrated in Fig. 2, we designed a 3D metacrystal with hexagonal lattice. The main body of the unit cell (Fig. 2a) is a staggered-ethane-shaped metallic particle possessing $D_{3d}$ point group symmetry. Fig. 2c calculates the bulk band structure of the metacrystal on different $k_z$ planes. In the results, the second and third bands intersect at $\bar{K}/\bar{K}'$ points, as predicted by the tight-binding model. Moreover, there exist two other redundant bands (the first and fourth bands) belonging to the other electromagnetic polarization (see the eigen mode profiles in Supplementary Note 2). As in the tight-binding model results, two Dirac points locate at the same frequency for $k_z = 0$ but have opposite frequency shifts for nonzero $k_z$. In 3D momentum space, the two Dirac cones extrude along the $k_z$ direction to form two ANLs (see Fig. 2d). In the designed metacrystal, the nodal line degeneracies are protected by the PT symmetry. Besides, they are pinned at BZ edges due to $C_3^z$ rotation symmetry.

To confirm the existence of bulk ANLs, we fabricate the designed metacrystal using printed circuit board (PCB) technology (see the photograph in Fig. 3a). The bulk ANLs are experimentally demonstrated by scanning the electromagnetic fields inside the bulk crystal. To measure the field inside, air holes are reserved at unit cell corners to plug in an antenna (inset of Fig. 3a). The purple line in Fig. 3a outlines the measured plane for field scanning. As illustrated in Fig. 3b, a source antenna is inserted from the bottom, whereas a probe antenna scans the $E_z$ fields on an $x$–$z$ plane inside the sample (purple rectangle). Since a half-wavelength dipole antenna (approximately a point source) was used in our experiment, modes with any $k_z$ component can be excited in principle. Thus, field profile for each $k_z$ can be extracted by Fourier transform. By performing Fourier transform on the measured field, we obtain the projected bulk bands onto the $k_x$-$k_z$ plane. Fig. 3i–m show the resulting Fourier spectra for different values of $k_z$. Compared with the corresponding calculated band structures in Fig. 3d–h, the in-plane

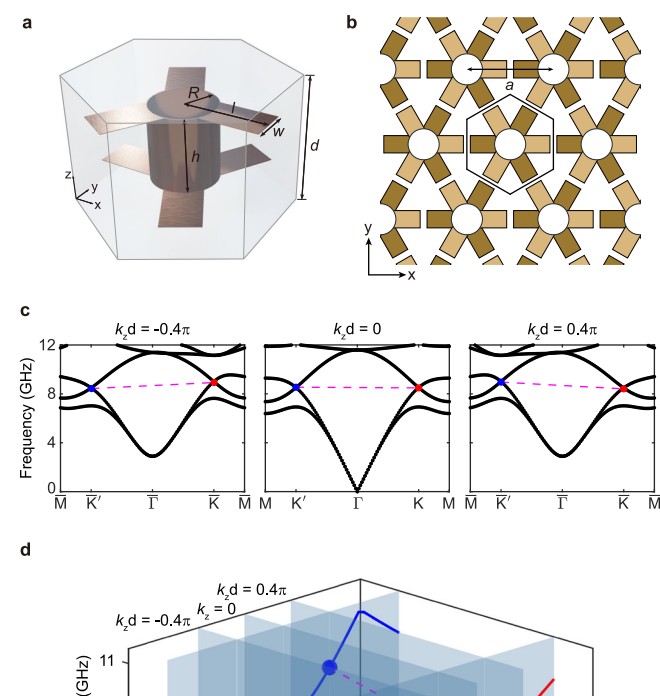

**Fig. 2 | Photonic metacrystal possessing ANLs.** The unit cell (**a**) and top view (**b**) of the photonic metacrystal with $D_{3d}$ point group symmetry. Structural parameters: $h = 4$ mm, $l = 4.6$ mm, $w = 2$ mm, $R = 1.8$ mm. The in-plane and out-of-plane lattice constants are $a = 10$ mm and $d = 7$ mm, respectively. **c** Bulk band structures on different $k_z$ planes. For the zero $k_z$, two Dirac points locate at the same frequency, resulting in a flat edge band (dashed magenta line); for nonzero $k_z$, the two Dirac points have opposite frequency shifts, yielding a tilted antichiral edge band. **d** ANLs constructed by aligning the Dirac points on different $k_z$ planes.

bulk dispersions, as well as the frequency shift of two Dirac points, evolve as $k_z$ varies. For a nonzero $k_z$ (e.g., $k_z = -0.4\pi/d$), the frequency shifts of Dirac points at $\bar{K}$ and $\bar{K}'$ differ. Consequently, we expect a tilted antichiral edge band pinned at two Dirac points. Notably, because our system is time-reversal-invariant, the frequency shifting condition is reverse for opposite $k_z$ (Fig. 3i, m). Our experimental results agree well with the calculate band structures, except for some additional signals at low frequency range caused by the transmission line modes between the source coaxial cable and probe coaxial cable.

## Experimental observation of the antichiral surface states
One direct manifestation of the topological features of the nodal line semimetals is the presence of topological surface states. In this study, we consider the zigzag surfaces (perpendicular to the $y$-axis) of the 3D hexagonal metacrystal. Both its upper ($+y$) surface and lower ($-y$) surface are bounded by metallic plates (see Fig. 4a). The corresponding surface BZ is shown in Fig. 4b, where two Dirac nodal lines are projected to $-\bar{H}\bar{H}$ and $-\bar{H}'\bar{H}'$. Guaranteed by the opposite Berry phases carried by two ANLs, a nontrivial surface band must fill inside or outside the two projected nodal lines (depending on the surface termination; see Supplementary Note 6). Because each 2D subsystem with fixed nonzero $k_z$ is analogous to an MHM, the surface band has a tilted dispersion connecting two Dirac points. As depicted in Fig. 4d–h,

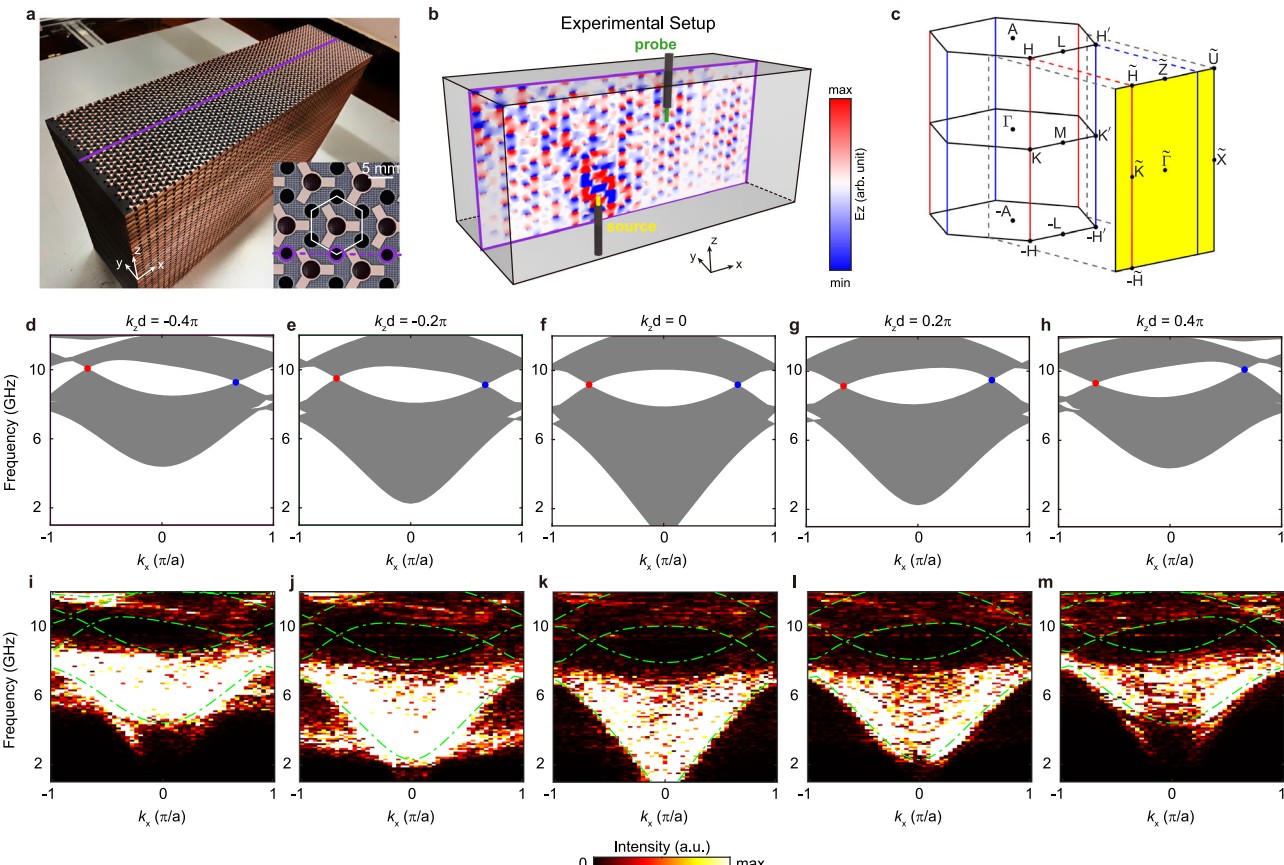

**Fig. 3 | Experimental observation of bulk ANLs. a** Photograph of the sample. The purple line depicts the measured plane to probe the bulk states. Inset: enlarged top view of the structure. Purple circles depict the reserved air holes for probing. **b** Experimental setup for probing the bulk band structure. A dipole source antenna (yellow) is inserted from the bottom, whereas the probe antenna (green) scans the middle $x$–$z$ plane hole-by-hole. A numerically simulated $E_z$ distribution at 9.5 GHz is plotted at the measured plane. **c** Bulk Brillouin zone and its surface projection

(yellow plane). The ANLs at $-HH$ and $-H'H'$ are projected to $-\bar{H}\bar{H}$ and $-\bar{H}'\bar{H}'$ of the surface BZ, respectively. **d–h** Simulated projected bulk bands on different $k_z$ planes. The blue and red dots denote the Dirac points at $\bar{K}$ and $\bar{K}'$ points, respectively. Because the system is time-reversal-invariant, the frequency shifting condition is reversed for opposite $k_z$. **i–m** Measured projected bulk bands. The green dot-dashed lines outline the simulated band edges.

the surface bands for the upper surface (blue dashed line) and lower surface (green dashed line) are degenerate due to a mirror reflection in the $y$ direction. Thus, the upper and lower surface modes propagate in the same direction, which is the hallmark feature of antichiral surface states. Besides the antichiral surface band pinned at Dirac points, another band appears at a higher frequency (the purple lines in Fig. 4d–h). Because it is not protected by the Berry phase of Dirac points, this band can be moved out of the gap by perturbation (see Supplementary Note 6). In the entire 2D surface BZ, the surface band looks like a ribbon with a longitudinal twist (Fig. 4c). Moreover, a recent study of an acoustic system proposed a nodal structure of straight nodal lines, supporting ribbon surface states that connect two straight nodal lines[55]. However, the two nodal lines have symmetric dispersions and do not exhibit antichiral surface states due to the $C_6$ symmetry of their system.

To the best of our knowledge, such type of unique twisted ribbon surface state has never been experimentally observed in any physical system. Fig. 4a illustrates the experimental setup to measure the surface dispersion. We measure the surface electric fields by plugging the probe antenna into a series of holes near the metallic plate. The source antenna (not shown in Fig. 4a) is placed near the sample surface to excite surface waves. The Fourier spectra of the scanned fields for the upper and lower surfaces are plotted in Fig. 4i–r. A tilted surface band connecting the two Dirac points can be clearly seen in our experimental data, which agrees well with the simulated results in Fig. 4d–h. Notably, the tilting directions (group velocity) of the surface bands are

locked to the $k_z$ component, suggesting a $k_z$-dependent one-way transport behavior.

## $k_z$-dependent antichiral surface transport

As mentioned in the tight-binding analysis, each reduced 2D system is analogous to an MHM with its local magnetic flux proportional to $k_z$. Consequently, the 2D subsystems with positive $k_z$ support two copropagating rightward surface modes compensated by leftward bulk modes (Fig. 5a), whereas those with negative $k_z$ support two copropagating leftward surface modes compensated by rightward bulk modes (Fig. 5g). Notably, these rightward or leftward modes would propagate in the $z$ direction in the meantime due to their nonzero $k_z$ components.

To visualize the $k_z$-dependent antichiral transport behavior, we simulate the intensity profile excited by two line sources and then extract certain $k_z$ components using Fourier transformation. Figure 5b–d depict the field profiles with $k_z = 0.2\pi/d$ at 9.5 GHz. Rightward surface waves are excited on both surfaces. For positive $k_z$ components (Fig. 5c, d), the excited rightward surface waves will shift a little in the $+z$ direction due to their group velocities in the $z$ direction. Similar results can be found in the field profiles for $k_z = -0.2\pi/d$ (Fig. h–j) but with an opposite propagation direction. Such a phenomenon was experimentally confirmed by the scanned electric field on the sample surface. After the Fourier transform, an excited field with a certain value of $k_z$ can be obtained. To characterize the unidirectionality of the antichiral surface mode, we define the directionality of the surface transport. For example, we obtain the electric field

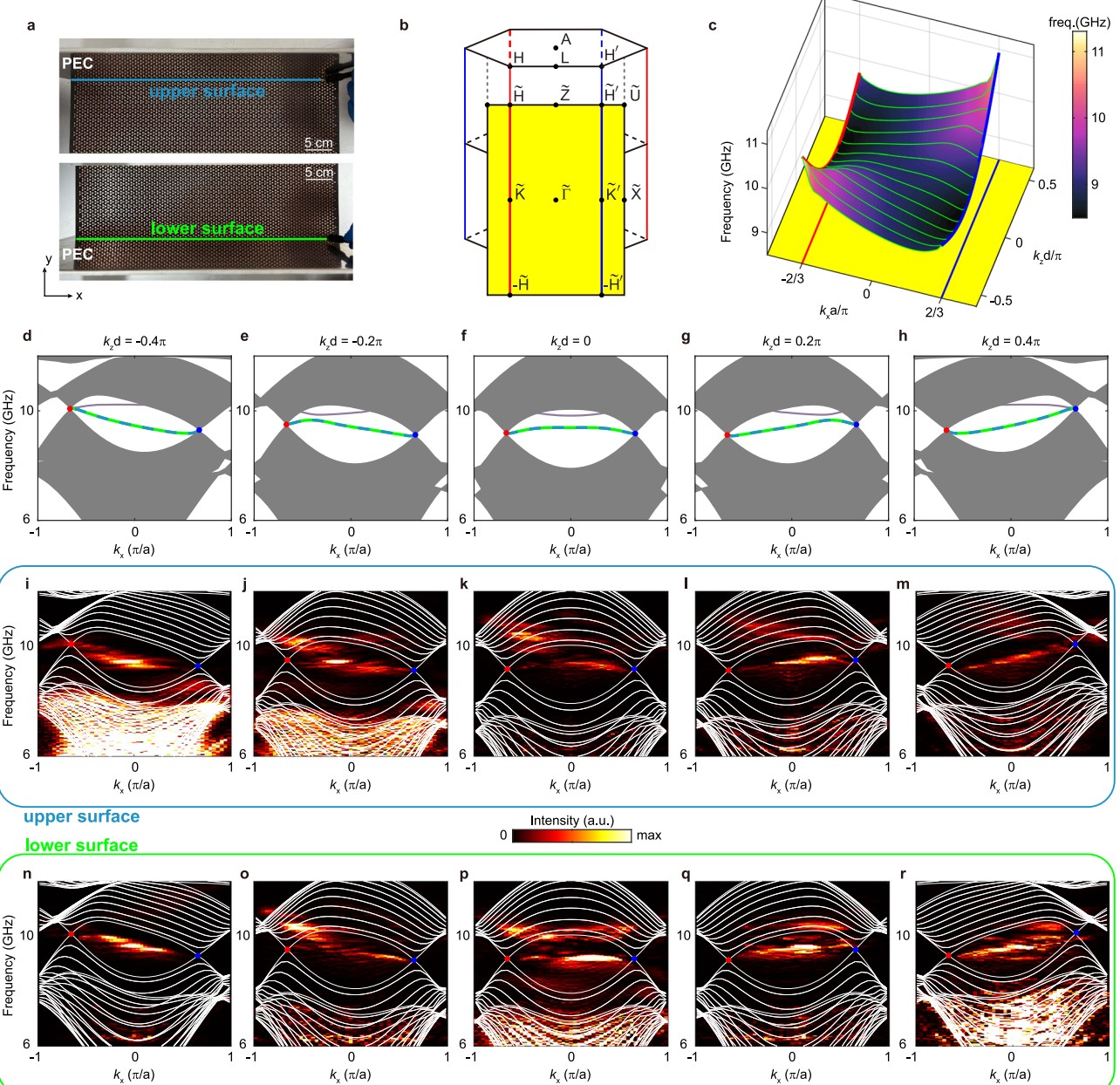

**Fig. 4 | Experimental observation of twisted ribbon surface state.**
**a** Experimental setup for probing the surface dispersion. The upper/lower surface is covered with an aluminum plate (deemed a PEC in the microwave regime). The probe antenna scans the crystal surface hole-by-hole. **b** Bulk Brillouin zone and its surface projection (yellow plane). **c** Simulated surface dispersion in the shape of a twisted ribbon. For clarity, the projected bulk bands are concealed, leaving only the surface bands. **d**–**h** Simulated projected band structure on different $k_z$ planes. The antichiral surface bands for the upper and lower surfaces are highlighted by the cyan and green dashed lines, respectively. The purple line denotes a trivial surface band that is not protected by the nontrivial Berry phase of Dirac point. **i**–**r** Measured surface dispersion. The white lines represent the simulated bulk bands.

---

with $k_z = 0.2\pi/d$ through Fourier transform and then calculate the energy ratio between the excited electric field on the right of the source and the field on the left of the source, namely

$$\eta = 10 \lg(|E_{z,Right}|^2/|E_{z,Left}|^2) \tag{5}$$

(details in Supplementary Note 8). Fig. 5e depicts the directionality spectrum obtained on the lower surface for $k_z = 0.2\pi/d$. The directionality is always positive in the frequency regime corresponding to the antichiral surface mode, meaning that most excited surface waves propagate to the right. Fig. 5k shows a similar result for $k_z = -0.2\pi/d$, except for the reversed propagation direction of surface waves.

Notably, the directionality spectra obtained on the upper surface (Fig. 5f, l) show similar $k_z$-dependent unidirectional surface transport, with some minor discrepancies. The reason may be the different excitation efficiencies on both surfaces due to sample imperfection.

Our results show that the surface modes with nonzero $k_z$ do copropagate along the top and bottom surfaces of a strip sample in the same direction, which is the hallmark of antichiral surface states. Further, the propagation directions of these surface states are locked to the $k_z$ component, indicating a unique $k_z$-dependent antichiral surface transport in our time-reversal metacrystals. More data of the $k_z$-dependent antichiral surface transport for different $k_z$ components can be found in Supplementary Note 9.

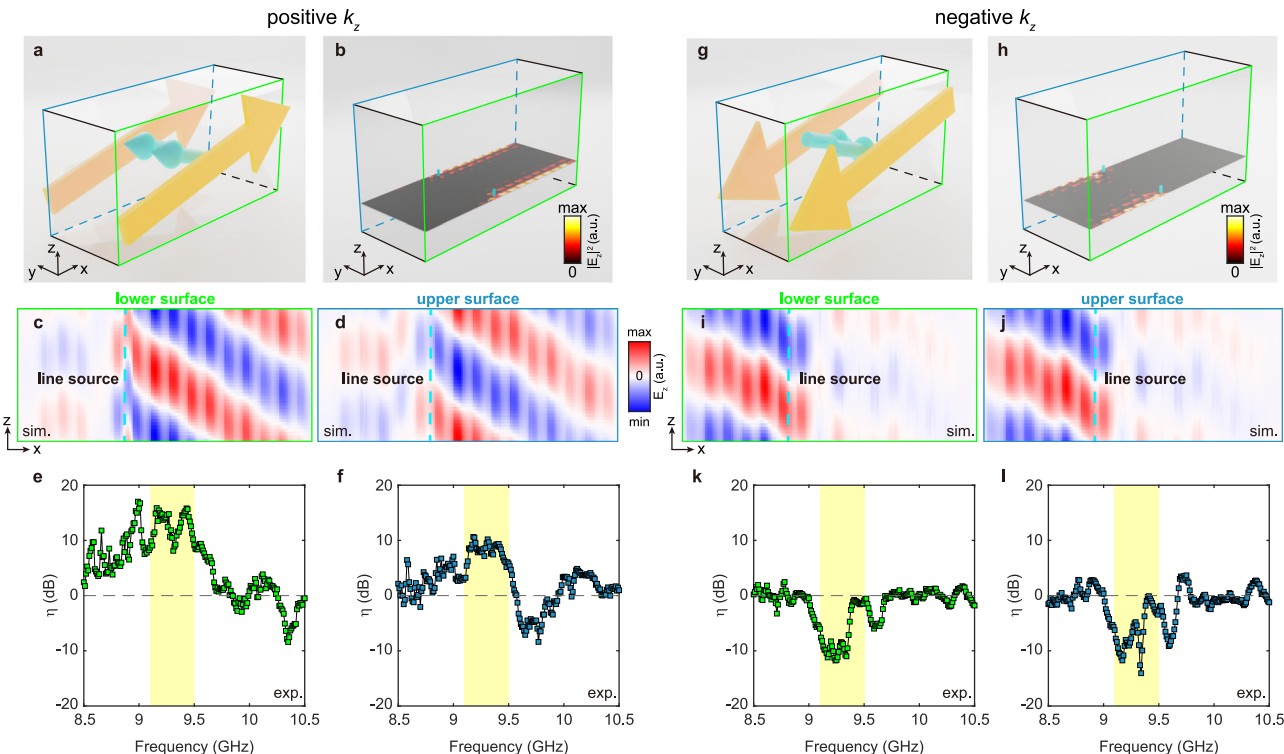

**Fig. 5 | $k_z$-dependent antichiral surface transport.** Sketch of the energy flow for bulk modes (cyan arrows) and surface modes (orange arrows) with positive $k_z$ (**a**) and negative $k_z$ (**g**). **b** Simulated intensity profiles for $k_z = 0.2\pi/d$ at 9.5 GHz. The cyan cylinders represent the line sources. On both surfaces, the electromagnetic waves with positive $k_z$ copropagate in the $+x$ direction. **c**, **d** Simulated $E_z$ profile on both surfaces for $k_z = 0.2\pi/d$. **e**, **f** Directionality spectra in the frequency regime of

the antichiral surface state for $k_z = 0.2\pi/d$. The frequency regime corresponding to the antichiral surface state is highlighted in yellow. **h** Simulated intensity profiles for $k_z = -0.2\pi/d$. On both surfaces, the electromagnetic waves with negative $k_z$ copropagate in the $-x$ direction. **i**, **j** Simulated $E_z$ electric field profile on both surfaces for $k_z = -0.2\pi/d$. **k**–**l** Directionality spectra for $k_z = -0.2\pi/d$.

## Discussion

In this letter, we realize antichiral surface states in a 3D time-reversal-invariant photonic metacrystal by introducing a synthetic gauge flux via dimension reduction. For each $k_z$, our 3D system reduces to a 2D subsystem analogous to an MHM and exhibits two frequency-shifted Dirac points. This indicates that our 3D metacrystal is a photonic semimetal with two straight nodal lines aligned in the $k_z$ direction. Unlike previous work on nodal line semimetals, the two bulk nodal lines in our system have asymmetric frequency dispersions, yielding a topological surface band in the shape of a twisted ribbon. It is the twisted ribbon surface dispersion that caused a $k_z$-dependent antichiral surface transport on the crystal boundary, in a time-reversal manner. The bulk ANLs and twisted ribbon surface band are experimentally confirmed by the Fourier spectra of the measured electric fields. The associated $k_z$-dependent antichiral surface transport is verified by simulated intensity distributions and measured transmission spectra. These unique $k_z$-dependent antichiral surface states extend the variety of unidirectional surface states other than chiral/helical/antichiral edge states. Our work also implies that the frequency/energy dimension does serve as an unexplored degree of freedom that may yield novel topological phases of matter. Although our idea is demonstrated in a photonic system, the results can be readily extended to other systems.

## Methods

### Numerical calculations

Commercial finite element software COMSOL Multiphysics was used in all full-wave simulations. The unit cell of our metacrystal is a 3D hexagon, of which we applied Floquet boundary conditions on the outer boundaries. The metallic structures were assumed to be perfect electric conductors (PECs) in calculations. In calculating the projected

band structure in Fig. 4, a metacrystal strip with 15 periods in the $y$-direction was considered and bounded by two PEC boundaries. To calculate the intensity profile in Fig. 5, a finite sample with $16 \times 5 \times 10$ periods was considered. After obtaining the electric field, we performed Fourier transform to extract the field profiles for different $k_z$ values.

### Samples

The photonic metacrystal in our work has an AA-stacked honeycomb structure. The oblique and top views of the unit cell are shown in Figs. 2a and 2b, respectively. A metallic particle satisfying $D_{3d}$ symmetry is embedded in a dielectric background with a relative permittivity of $\varepsilon_r = 2.65$. The staggered-ethane-shaped particle has a 4-mm height and a 1.8-mm radius. Its six metallic arms are 4.6 mm long and 2 mm wide. In-plane and out-of-plane lattice constants are $a = 10$ mm and $d = 7$ mm, respectively. We fabricated an experimental sample using print circuit boards (PCBs) technology. The 3D metacrystal was constructed by stacking the patterned PCBs and dielectric spacers alternately in the $z$ direction. The thicknesses of the PCBs and the dielectric spacers are 4 mm and 3 mm. The sample has 50 periods in the $x$ direction, 8 periods in the $y$ direction and 40 periods in the $z$ direction.

To probe the surface modes, we bounded the metacrystal in the $y$ direction (the zigzag surface) with an aluminum plate (Fig. 4a). In order to probe the electromagnetic field inside the sample, we reserved holes at all six corners of each hexagonal cell (see inset of Fig. 3a) for sticking into with the probe antenna.

### Experimental setup

In our microwave experiment, a vector network analyzer (KeysightE5071C) was used to excite and probe the electromagnetic waves

in the sample. The length of antenna was optimized to maintain considerable radiation efficiency. Considering the frequency variation of the offset Dirac points from 8 GHz to 10 GHz, we used a 6-mm-long monopole antenna in the measurement. The measured spectra ranged from 1 GHz to 15 GHz with a resolution of 0.01 GHz.

The experimental setup for the bulk mode measurement is shown in Fig. 3b. A source antenna was inserted from bottom of the sample. In addition, a probe antenna was stuck into the sample from the reserved air holes at unit cell corners (purple lines in Fig. 3a) to probe the electromagnetic field. To obtain the projected bulk bands in Fig. 3, we scanned the middle $x-z$ plane in the reserved holes hole-by-hole. The projected bulk bands on different $k_z$ planes were obtained by performing Fourier transform on the measured $E_z$ field. The experimental setup for measuring the surface dispersion in Fig. 4 is similar except that a metallic plate was added beside the metacrystal (Fig. 4a). In this case, the source antenna is fixed at the sample surface, whereas the probe antenna scans the surface hole-by-hole. The measured surface transmission in Fig. 5 was obtained by extracting the $E_z$ field with a specific $k_z$ by Fourier transforming the $E_z$ field in each hole.

## Data availability
All data needed to evaluate the conclusions in the paper are present in the paper and/or the Supplementary Information. Additional data related to this paper may be requested from the corresponding authors.

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

## Acknowledgements

This work was supported by National Natural Science Foundation of China (Grant Nos. 62035016, 12074443), Guangdong Basic and Applied Basic Research Foundation (Grant No. 2019B151502036), Guangzhou Basic and Applied Basic Research Foundation (Grant No. 202102080414), Fundamental Research Funds for the Central Universities (Grant No. 22qntd3001).

## Author contributions

W.J.C. and J.W.D. supervised this project; J.W.L. conceived the origin idea, performed the numerical simulation and analytical analysis with the help of W.J.C.; J.W.L. performed the experiments and analyzed the measurement with the help of F.L.S., W.J.C., X.D.C., K.S. and K.C.; J.W.L. and W.J.C. wrote this manuscript and all authors contributed to scientific discussion about this manuscript.

## Competing interests

The authors declare no competing interests.
