## [Peer review file · Nature Communications]

REVIEWER COMMENTS

Reviewer #1 (Remarks to the Author):

The authors have designed, analyzed, and measured a photonic metacrystal that supports the so-called anti-chiral surface waves, which are two waves that co-propagate rather than counter-propagating as with typical chiral waves. Apparently such states normally only occur in systems without time reversal symmetry, but the authors have obtained them in a time reversal symmetric system by using k_z as an “artificial gauge field”. This approach has been used in a variety of other photonic systems, such as coupled optical fibers, to achieve an artificial breaking of time reversal symmetry, and it is appropriate here as well. The authors design this crystal using printed circuit board technology, and they measure it with a pair of probe antennas. They do indeed find surface waves that propagate at an angle across the crystal, and of course the waves propagate in the opposite angle in the reverse k_z direction, since the overall structure is reciprocal. The unique aspect, and apparently the defining feature of “anti-chiral” waves is that a second surface wave also propagates in the same direction, although at a different frequency. The authors do extensive analysis using a tight binding model as well as electromagnetic simulations to identify the shifted Dirac cones that are used to explain this effect. Overall the work is a high-quality, and the authors have two models that explain the results and are consistent with their experimental data. The authors describe the band structure as having a twisted ribbon shape, which apparently refers to the saddle-point type shape. All of the work is interesting, although it is somewhat unclear what the impact will be beyond validating some of the existing theories on such states in a photonic system. Nonetheless, this is the first such demonstration. The only significant comment I have is that the English could be improved in some areas of the paper.

Reviewer #2 (Remarks to the Author):

The authors develop a novel design to realize antichiral surface states in a three-dimensional photonic metacrystal in a time-invariant manner for the first time. I think this work has the novelty and impact required by Nature Communications, but only when the below issues have been clarified I may provide a definite recommendation.

1. Whether such k_z -dependent antichiral surface states are strong robust against the imperfections on the transport path.
2. How does the tight-binding model correspond to the designed 3D photonic crystal? How is the coupling between layers regulated in photonic crystal?
3. The authors mentioned “Guaranteed by the opposite Berry phases carried by two ANLs, a nontrivial surface band must fill inside or outside two projected nodal lines”. I want to know whether the surface

states are always locked between two Dirac nodes and whether this phenomenon can be described by a certain topological invariant.

4. Are these surface states sensitive to geometric parameters? Does the varying width between the photonic crystal and the metal plate affect the dispersion relations or propagation behaviors of the surface states?

5. In Figs. 3 and 4, it is seen that the simulation and experimental results of surface states do not seem to be in good agreement, what is the main reason?

6. How does the source lock $+k_z$ or $-k_z$? It is suggested that the author should give more detailed information in simulations and experiments.

7. There exist two green dispersion curves in Figs. 4c-4g, are they both topologically protected surface states? What are the physical origins of them?

8. To characterize the unidirectionality of surface states, the authors define the directionality as the energy ratio between the excited electric field on the right of the source and the field on the left. I want to know how the author obtain these electric fields. Is it the integral of a line or a region electric field? The more detailed calculation information should be described. Besides, to show the one-way transmission more intuitively, I suggest that the author should demonstrate the electric field distributions in the x - z plane.

9. In Figs. 5g-5j, the directionality spectra of $+k_x$ and $-k_x$ surface states are not very agreement, what is the main reason?

10. The authors mentioned "due to the intrinsic material loss and weak magneto-optic effect of gyromagnetic materials, this scheme can hardly be extended to optical regime". Then, I wonder if the scheme proposed here can be extended to the optical regime?

Reviewer #3 (Remarks to the Author):

The article by Jian-Wei Liu and co-workers entitled "Antichiral

surface states in time reversal photonic semimetals" shows the realization of antichiral surface states in a time-invariant manner and demonstrates their idea within a three-dimensional photonic metal crystal.

Anti-chiral edge state was firstly proposed in the modified Haldane model with breaking time-reversal symmetry, where two degenerate edge states (corresponding to the top and bottom edges) bridge two Dirac cones. Because the two Dirac cones sit on different energies, thus the two degenerate edge states exhibit unidirectional transport. The authors proposed a photonic system possessing two asymmetrically dispersed Dirac nodal lines. At some nonzero k_z , the three-dimensional system can be regarded as a 2D

modified Haldane model, where two Dirac cones locate on different energies. Finally, they observe the k_z -dependent antichiral surface states, or twisted ribbon surface states, considering different k_z .

The main claim of the manuscript is to have experimentally realized time-reversal invariant anti-chiral surface states.

Introducing k_z (or dimension reduction) to realize the time-reversal breaking 2D system has been widely used, such as [Nature Physics 11, 920–924 (2015)]. However, there are no antichiral edge states reported in 3D nodal line systems so far. Here the authors cut nodal lines to observe the antichiral edge states, which correspond to the usual drumhead surface states in the nodal line system.

The theoretical proposal is interesting and the experimental results are convincing. Therefore, the work can be recommended for publication on Nature Communications after the following comments and suggestions are well addressed.

1- The experimental results shown in Fig. 3 are not clear enough, the authors may need to make it clearer by trying different experimental setups, i.e. changing the source excitation positions. The authors can also try to weaken the simulation results (I guess making the simulated results thinner or dashed) to reveal the experimental results more.

2- For conventional antichiral edge states, the top and bottom edge states are degenerate, however, they are not in this work (see Fig. 4). They even do not terminate at the projected Dirac points. The author may need to discuss this. Furthermore, the authors may need to use different colours to plot the different surface states.

3- The out-of-plane lattice constant d is missing in Fig. 2a, and the metallic structure's height is missing.

Reviewer #4 (Remarks to the Author):

The authors present the theoretical and experimental investigation of the emergence of antichiral surface states in photonic crystals system invariant to time-reversal. The system consists of a 3D photonic crystal with a z -asymmetric unit cell and rotational symmetry. For the theoretical analysis they employ a two dimensional reduction of the 3D Hamiltonian describing the crystal by introducing an intercoupling coefficient. For the experimental realization they use a microwave crystal fabricated by

standard PCB technology and vector analyzer. The work is generally relevant however several points need substantial improvement before considering for publication. My comments are:

1. The surface states under consideration are named topological. Could the Authors provide a calculation of a topological charge to justify the use of the term?
2. It is a bit difficult to understand from which is the plane of the investigations in Fig. 3 and 4. The Authors refer to the center of the bulk crystals. Is it the same as the center of the cell? An image of the cross-section would help a lot.
3. How many points do the measurements along the x-z plan include? Could the authors provide the field data before the FFT, indicatively, for some frequencies?
4. There are some significant qualitative differences in the simulated and experimental data for Figure 3. For example in Fig.3j there seems to be an additional band around 3Ghz which is not predicted in theory. Are Fig.3i and Fig.3m supposed to be mirrored? Why is it so different in experiment? In Figure 4k, the symmetric case, there is an asymmetry in the high frequencies. Could the Authors comment on that?
5. The authors mention that “we consider the zigzag surface of the 3D hexagonal metacrystal by placing a metallic plate next to the sample”. This needs to be further clarified and discussed in the manuscript. What does this mirror symmetry imposed in the electromagnetic fields? Which is the termination of the photonic crystal before the PEC wall selected for the case appearing in figure 4?
6. Is the probe antenna in Figure 4 placed in between the PC wall and the sample?
7. Is the BZ presented in Fig3c as it is mentioned in Line 162 or in Fig4c
8. Regarding the surface transport, it hard to understand where the surface mode propagates in the simulation. Is a 3D system considered in the simulation? Is the cross-section of the field in Fig.5 assumed just above the structure?
9. It would be interesting if the authors showed the scanned electric field data before the FFT in Figure 5.
10. It would be better if the term time reversal invariant would be appearing in the title.
11. The quality of the figures needs to be improved.
12. A careful proof reading is necessary.

Response to Reviewer #1

Comment: The authors have designed, analyze, and measured a photonic metacrystal that supports the so-called anti-chiral surface waves, which are two waves that co-propagate rather than counter-propagating as with typical chiral waves. Apparently such states normally only occur in systems without time reversal symmetry, but the authors have obtained them in a time reversal symmetric system by using k_z as an “artificial gauge field”. This approach has been used in a variety of other photonic systems, such as coupled optical fibers, to achieve an artificial breaking of time reversal symmetry, and it is appropriate here as well. The authors design this crystal using printed circuit board technology, and they measure it with a pair of probe antennas. They do indeed find surface waves that propagate at an angle across the crystal, and of course the waves propagate in the opposite angle in the reverse k_z direction, since the overall structure is reciprocal. The unique aspect, and apparently the defining feature of “anti-chiral” waves is that a second surface wave also propagates in the same direction, although at a different frequency. The authors do extensive analysis using a tight binding model as well as electromagnetic simulations to identify the shifted Dirac cones that are used to explain this effect. Overall the work is a high-quality, and the others have two models that explain the results and are consistent with their experimental data. The authors describe the band structure as having a twisted ribbon shape, which apparently refers to the saddle-point type shape. All of the work is interesting, although it is somewhat unclear what the impact will be beyond validating some of the existing theories on such states in a photonic system. Nonetheless, this is the first such demonstration. The only significant comment I have is that the English could be improved in some areas of the paper.

Response:

We thank the reviewer for his/her careful reading of our manuscript and the positive comments. In the following, we have addressed all the comments and revised the manuscript accordingly.

(1) In our manuscript, we are inclined to describe the surface dispersion as having a twisted ribbon shape, in order to emphasize its remarkable feature. To see this, we calculate the surface dispersion using tight-binding model in Fig. S4. Because the surface band occupies a rectangular area in the surface Brillouin zone, its overall shape looks like a ribbon bounded by two straight nodal lines. Because of the asymmetric dispersions of the two nodal lines, the ribbon is tilted to the left in positive k_z regime while tilted to the right in negative k_z regime, leading to a “twisted ribbon”. This is a distinctive feature for this type of TR-invariant antichiral surface states. On the other hand, we can see that the surface band is somewhat a convex function in the k_x direction, not a straight line pinned at two DPs, making itself a saddle point shape. But this saddle point shape is not a deterministic feature of the sTR-invariant antichiral surface states. To avoid misunderstanding, relevant descriptions have been modified accordingly in the revised manuscript (Line 122-126).

Fig. S4 Twisted ribbon surface dispersion calculated by tight-binding model. Two asymmetric nodal lines are depicted in red and blue, respectively.

(2) In this work, we realize antichiral surface transport by utilizing a synthetic gauge flux. From this perspective, we just validate the existing “antichiral edge transport” using another approach. But remember that our approach needs not engage TR breaking, which is unfavorable in photonic systems, for the intrinsic loss and weak response of magneto-optical materials. Furthermore, our results turn out that the original antichiral edge states in 2D systems have a 3D counterpart in nodal line semimetals. This has never been proposed and demonstrated before, to the best of our knowledge. We believe that our findings will enrich the ways to mold the flow of light, although photonic devices based on this mechanism remain untamed. Finally, these ideas are not constrained to photonic context and can be easily extended to other systems and may inspire potential applications, such as acoustic devices.

(3) We thank the referee for this suggestion. We have carefully read and revised our manuscript with the help of native speaker.

Responses to Reviewer #2

Comment: The authors develop a novel design to realize antichiral surface states in a three-dimensional photonic metacrystal in a time-invariant manner for the first time. I think this work has the novelty and impact required by Nature Communications, but only when the below issues have been clarified I may provide a definite recommendation.

Response: We thank the reviewer for his/her careful reading of our manuscript and the positive comments. In the following, we have addressed all the comments and revised the manuscript accordingly.

Comment 1: Whether such k_z -dependent antichiral surface states are strong robust against the imperfections on the transport path.

Response:

Since the surface band bridging two DPs is tilted for nonzero k_z , the crystal surface should support unidirectional transport. However, for the absence of a completed band gap, some of the surface waves may be scattered into the bulk crystal, which is similar to the 2D antichiral edge states. To confirm this, we simulate two cases of surface imperfection (PEC block and Ω -shaped bending) in Fig. S8. In both cases, most of the waves keep propagating rightward on the surface, despite some energy losses (scattered into the bulk).

Fig. S8 **Simulated antichiral surface state with positive k_z .** **a** Schematic of surface perturbation. The lower surface is perturbed by a metallic block, while the upper surface has no imperfection on the transport path. **b** Intensity profile of the excited surface states. **c** Schematic of surface configuration with one straight path and one Ω path. **d** Antichiral surface transport on a flat surface and an Ω -shaped surface.

Comment 2: How does the tight-binding model correspond to the designed 3D photonic crystal? How is the coupling between layers regulated in photonic crystal?

Response:

As we know, tight-binding model is a good approximation in the calculation of electronic band structure, because electronic wave functions are usually well localized near the atoms. However, this assumption breaks down in photonic systems, because EM wave can propagate, rather than decay, in the background medium (say, the dielectric medium surrounding the meta-atoms in Fig. 2a). The multiple scattering of EM waves in photonic crystals is rather complicated and cannot be solved analytically (especially for a 3D structure). Rigorous photonic band structures are usually given by numerical calculations.

In designing our metacrystal, tight-binding model cannot give us the detailed

structural parameters, but it can guide us to the appropriate space group symmetry required by ANLs. In the band structure of a crystal, most of the band degeneracies are guaranteed by crystalline symmetry (for example, Dirac cones can always be found in C_{6v} systems). Therefore, we believe a photonic crystal belonging to $\overline{P31m}$ group should host two ANLs pinned at BZ edges. This is verified in our simulation results.

The strength of interlayer coupling in our photonic metacrystal can be controlled by the thickness of the dielectric spacer between meta-atoms.

The above discussions have been added in the revised manuscript (Line 132-138).

Comment 3: The authors mentioned “Guaranteed by the opposite Berry phases carried by two ANLs, a nontrivial surface band must fill inside or outside two projected nodal lines”. I want to know whether the surface states are always locked between two Dirac nodes and whether this phenomenon can be described by a certain topological invariant.

Response:

(1) The antichiral surface bands are always locked between two DPs. To verify this property, we calculate the surface dispersions for different surface truncations in Supplementary Note 6. As shown in Fig. S11, the change of surface truncation does affect the specific surface dispersion, but the surface bands are always pinned at two DPs.

(2) The existence of such surface band is related to the Zak phase of the crystal. In Supplementary Note 3, we calculate Zak phases accumulated in the k_y direction, as the crystal is truncated in the y direction. In the surface BZ, the Zak phase is π inside two projected NLs while it is 0 outside. The sudden jump of the Zak phase is due to the π Berry phase carried by the DNLs. Interestingly, the topological feature of the DNLs is also manifested in a reflection phase winding near DPs [Deng, et al., Light: Sci. Appl. 11:134 (2022)]. This property is numerically verified in our metacrystal (see the response to Comment 7 below). Since the existence of surface state should satisfy the relation of $\phi_{PC} + \phi_{PEC} = 2N\pi$, this 2π -winding can ensure at least one surface band emerged from DP, no matter the reflection phase of the other crystal.

Fig. S7 **a** Bulk band structure on the plane of $k_z = 0.2\pi/d$. **b** Distribution of Zak phase sum ($\theta_1 + \theta_2$) accumulated in the k_y direction, when $k_z = 0.2\pi/d$. **c** Zak phase distribution in the whole surface Brillouin zone.

Comment 4: Are these surface states sensitive to geometric parameters? Does the varying width between the photonic crystal and the metal plate affect the dispersion relations or propagation behaviors of the surface states?

Response:

The surface band dispersion is sensitive to geometric parameters of the crystal, such as the truncation position and the air gap width, see Figs. S11 & S12. However, due to the reflection phase winding near DP, there always exists at least one band stretched from the DP.

Comment 5: In Figs. 3 and 4, it is seen that the simulation and experimental results of surface states do not seem to be in good agreement, what is the main reason?

Response:

(1) In Fig. 3, there are some differences between the simulation and experimental results at around 3.5 GHz, especially for the cases with negative k_z (Figs. 3i&3j). We find that these additional signals come from the transmission line modes between the source coaxial cable and probe coaxial cable. Figure R1a below plots the measured field at 3.5 GHz, where we can see that most of the signals are localized near the cable connecting source antenna. Corresponding FFT spectra are shown in Fig. R1c.

To show this, we perform a Fourier transform on the electric field near the source antenna (dashed rectangle in Fig. R1b). Corresponding Fourier spectra (Fig. R1d) coincide with the additional signals around 3.5 GHz in Figs. 3i & 3j.

(2) In order to demonstrate antichiral surface dispersion clearer, we have carefully improved our experimental setup and updated the results in Fig. 4. In the new Fig. 4, the measured surface dispersions for both surfaces are shown. The projected bulk bands are highlighted with white lines and the Dirac points are highlighted with red and blue dots. In these new results, the surface bands can be clearly seen, connecting the two Dirac points.

Fig. R1 **a** Measured E_z field profile at 3.5GHz before FFT. **b** E_z field profile near the source antenna. Only the field in the region highlighted in dashed rectangle is reserved while field in the other region is set to be zero. **c** Fourier spectrum corresponding to original field data with $k_z = -0.2\pi/d$. **d** Fourier spectrum corresponding to the transmission line modes with $k_z = -0.2\pi/d$.

Comment 6: How does the source lock $+k_z$ or $-k_z$? It is suggested that the author should give more detailed information in simulations and experiments.

Response:

The source used in our experiment is a half-wavelength dipole antenna (approximately a point source), which can in principle have all k_z components. We measure the electric field on an x - z plane and then perform Fourier transform on the data to extract the field profile for each k_z , as shown in Fig. S15. Similarly, the numerical field profiles with different k_z (Figs. 5 & Fig. S15) are also extracted using Fourier transform.

To avoid misunderstanding, we have added more detailed information about data possessing in the manuscript, Line 158-160. The experimental setup and data possessing are also demonstrated in Method.

Comment 7: There exist two green dispersion curves in Figs. 4c-4g, are they both topologically protected surface states? What are the physical origins of them?

Response:

(1) To demonstrate antichiral surface dispersion clearer, we have carefully improved our experimental setup and updated the results in Fig. 4. As a consequence of the mirror reflection in the y direction, all the surface bands in Figs. 4d-h are doubly degenerate (one for the upper surface and the other for the lower surface). Apart from the antichiral surface band pinned at DPs (blue and green dashed lines), another surface band appears in higher frequency (purple lines). However, this double band is not protected by Berry phase of DPs and can be moved out of the gap by perturbation (see Supplementary Note 6).

(2) In previous work, the physical origin of this mid-gap surface states or drumhead surface states was studied via the reflection phase in the vicinity of DP [Deng, et al., Light: Sci. Appl. 11:134 (2022)]. To get insight into the physical origins of surface bands in our system, we similarly calculate the reflection phase of our metacrystal near the DP (see Supplementary Note 5). Due to the 2π -winding of reflection phase around DP, at least one surface band satisfying the resonance condition would emerge from the DP. This surface band is topological protected and is always pinned at the DP. Additionally, surface bands not pinned at the DP can also exist in the bulk gap in some circumstances (purple lines in Figs. 4d-h). However, these surface bands are not topologically protected.

Fig. 4 Experimental observation of twisted ribbon surface state. **a** Experimental setup for probing the surface dispersion. The upper/lower surface is covered with an aluminum plate (deemed a PEC in the microwave regime). The probe antenna scans the crystal surface hole-by-hole. **b** Bulk Brillouin zone and its surface projection (yellow plane). **c** Simulated surface dispersion in the shape of a twisted ribbon. For clarity, the projected bulk bands are concealed, leaving only the surface bands. **d-h** Simulated projected band structure on different k_z planes. The antichiral surface bands for the upper and lower surfaces are highlighted by the cyan and green dashed lines, respectively. The purple line denotes a trivial surface band that is not protected by the nontrivial Berry phase of Dirac point. **i-r** Measured surface dispersion. The white lines represent the simulated bulk bands.

Fig. S9 Reflection phase near Dirac point. Top view (a) and side view (b) of the setting to calculate the reflection phase. c Projected bulk band near the Dirac point with k_z equals zero. d-e Reflection phase at $k_x = 0.6\pi/a$ and $k_x = 0.7\pi/a$.

Comment 8: To characterize the unidirectionality of surface states, the authors define the directionality as the energy ratio between the excited electric field on the right of the source and the field on the left. I want to know how the author obtain these electric fields. Is it the integral of a line or a region electric field? The more detailed calculation information should be described. Besides, to show the one-way transmission more intuitively, I suggest that the author should demonstrate the electric field distributions in the x-z plane.

Response:

We thank the reviewer for his/her suggestion. To obtain the directionality, we first perform Fourier transform on the original measured field and then extract the k components with certain value of k_z to reconstruct the field profile, as shown in Fig. S14b. Based on the reconstructed profile, we integrate the electric field in the region on the left/right side of the source (dashed rectangles in Fig. S14b). Finally, the directionality is obtained as the energy ratio between two regions.

To demonstrate the one-way transmission in a more intuitive way, we have added a few indicative field profiles for different k_z in Fig. S15 and Supplementary Note 9. Although the electric fields on the right are obviously stronger than the ones on the left (indicating a unidirectional transport on the surface), the rightward propagating surface waves gradually decay in the x direction. The reason may be the imperfection of our sample. Since the sample is constructed by stacking PCBs in the z direction. The PCBs' out-of-flatness would lead to air gaps between neighboring boards and deteriorate the periodicity in the z direction. Therefore, the excited surface waves with certain k_z could

be scattered to the bulk states with other k_z , resulting in the attenuation of the antichiral surface waves.

To avoid misunderstanding, we have added more detailed information about data possessing in the manuscript, Line 218-224.

Fig. S14 Data processing for calculating the directionality of surface transport.

Fig. S15 Antichiral surface transport with different k_z component at different surfaces.

Comment 9: In Figs. 5g-5j, the directionality spectra of $+k_x$ and $-k_x$ surface states are not very agreement, what is the main reason?

Response:

In the revised manuscript, we have improved the results in Fig. 5. In the updated results of directionality spectra (Figs. 5e-f and 5k-l), there are still some minor differences for different surfaces. The reason may be the different excitation efficiencies on both surfaces due to the sample imperfection. On the other hand, the directionality spectra with $+k_z$ and $-k_z$ components are also different. The main reason may be the strong signals from the transmission line modes between antennas and PEC boundary.

Comment 10: The authors mentioned “due to the intrinsic material loss and weak magneto-optic effect of gyromagnetic materials, this scheme can hardly be extended to optical regime”. Then, I wonder if the scheme proposed here can be extended to the optical regime?

Response:

As we have discussed in the manuscript and in the response to Comment 2, we designed our 3D photonic crystal according to the space point group symmetry of the tight-binding model, i.e. $P\bar{3}1m$ (no. 162). It is such symmetry that ensures the existence of the asymmetric nodal lines and the resulting antichiral surface states. Therefore, we believe that our scheme can be extended to the optical regime as long as we design a micro-nano scale dielectric 3D photonic crystal possessing the same spatial symmetry.

Responses to Reviewer #3

Comment: The article by Jian-Wei Liu and co-workers entitled "Antichiral surface states in time reversal photonic semimetals" shows the realization of antichiral surface states in a time-invariant manner and demonstrates their idea within a three-dimensional photonic metal crystal.

Anti-chiral edge state was firstly proposed in the modified Haldane model with breaking time-reversal symmetry, where two degenerate edge states (corresponding to the top and bottom edges) bridge two Dirac cones. Because the two Dirac cones sit on different energies, thus the two degenerate edge states exhibit unidirectional transport. The authors proposed a photonic system possessing two asymmetrically dispersed Dirac nodal lines. At some nonzero k_z , the three-dimensional system can be regarded as a 2D modified Haldane model, where two Dirac cones locate on different energies. Finally, they observe the k_z -dependent antichiral surface states, or twisted ribbon surface states, considering different k_z .

The main claim of the manuscript is to have experimentally realized time-reversal invariant anti-chiral surface states.

Introducing k_z (or dimension reduction) to realize the time-reversal breaking 2D system has been widely used, such as [Nature Physics 11, 920–924 (2015)]. However, there are no antichiral edge states reported in 3D nodal line systems so far. Here the authors cut nodal lines to observe the antichiral edge states, which correspond to the usual drumhead surface states in the nodal line system.

The theoretical proposal is interesting and the experimental results are convincing. Therefore, the work can be recommended for publication on Nature Communications after the following comments and suggestions are well addressed.

Response:

We thank the reviewer for his/her careful reading of our manuscript and the positive comments. In the following, we have addressed all the comments and revised the manuscript accordingly.

Comment 1: The experimental results shown in Fig. 3 are not clear enough, the authors may need to make it clearer by trying different experimental setups, i.e. changing the source excitation positions. The authors can also try to weaken the simulation results (I guess making the simulated results thinner or dashed) to reveal the experimental results more.

Response:

We thank the referee for this helpful suggestion. In the new experimental data, we put the source antenna near the crystal surface, instead of plugging it into the bulk. This can effectively suppress the strong signals of transmission line modes. In the new Fig. 4, the antichiral surface bands can now be clearly seen.

To reveal the experimental data explicitly, we have made the simulated bulk bands in Fig. 3 thinner and dashed.

Comment 2: For conventional antichiral edge states, the top and bottom edge states are degenerate, however, they are not in this work (see Fig. 4). They even do not terminate at the projected Dirac points. The author may need to discuss this. Furthermore, the authors may need to use different colours to plot the different surface states.

Response:

We thank the referee for this comment. In this work, the upper surface and the lower surface have an identical dispersion, as a consequence of mirror reflection in the y direction. Thus, all the surface bands in the simulation of Figs. 4d-h are doubly degenerate (one for the upper surface and the other for the lower surface).

Furthermore, there exist two surface bands in these results. The one with lower frequency is pinned at two DPs while the other with higher frequency is not. This higher frequency band is not protected by the Berry phase of the DP and can be moved outside the gap by perturbation. To avoid misunderstanding, we have revised the figure captions and use different colors to distinguish the trivial/nontrivial surface bands.

In experiment, we remeasure the data separately for the upper surface and the lower surface in the new Figs. 4i-r. Both surfaces exhibit the same k_z -dependent antichiral surface bands, demonstrating their identical dispersion.

Fig. 4 Experimental observation of twisted ribbon surface state. **a** Experimental setup for probing the surface dispersion. The upper/lower surface is covered with an aluminum plate (deemed a PEC in the microwave regime). The probe antenna scans the crystal surface hole-by-hole. **b** Bulk Brillouin zone and its surface projection (yellow plane). **c** Simulated surface dispersion in the shape of a twisted ribbon. For clarity, the projected bulk bands are concealed, leaving only the surface bands. **d-h** Simulated projected band structure on different k_z planes. The antichiral surface bands for the upper and lower surfaces are highlighted by the cyan and green dashed lines, respectively. The purple line denotes a trivial surface band that is not protected by the nontrivial Berry phase of Dirac point. **i-r** Measured surface dispersion. The white lines represent the simulated bulk bands.

Comment 3: The out-of-plane lattice constant d is missing in Fig. 2a, and the metallic structure's height is missing.

Response:

We thank the reviewer for this helpful suggestion. Accordingly, we have revised Fig. 2a in the manuscript.

Fig. 2 Photonic metacrystal possessing ANLs. The unit cell (a) and top view (b) of the photonic metacrystal with D_{3d} point group symmetry. Structural parameters: $h = 4$ mm, $l = 4.6$ mm, $w = 2$ mm, $R = 1.8$ mm. The in-plane and out-of-plane lattice constants are $a = 10$ mm and $d = 7$ mm, respectively. **C** Bulk band structures on different k_z planes. For the zero k_z , two Dirac points locate at the same frequency, resulting in a flat edge band (dashed magenta line); for nonzero k_z , the two Dirac points have opposite frequency shifts, yielding a tilted antichiral edge band. **d** ANLs constructed by aligning the Dirac points on different k_z planes.

Responses to Reviewer #4

Comment: The authors present the theoretical and experimental investigation of the emergence of antichiral surface states in photonic crystals system invariant to time-

reversal. The system consists of a 3D photonic crystal with a z-asymmetric unit cell and rotational symmetry. For the theoretical analysis they employ a two dimensional reduction of the 3D Hamiltonian describing the crystal by introducing an intercoupling coefficient. For the experimental realization they use a microwave crystal fabricated by standard PCB technology and vector analyzer. The work is generally relevant however several points need substantial improvement before considering for publication.

Response:

We thank the reviewer for his/her careful reading of our manuscript. In the following, we have addressed all the comments/suggestions below point-by-point, and revised manuscript accordingly.

Comment 1: The surface states under consideration are named topological. Could the Authors provide a calculation of a topological charge to justify the use of the term?

Response:

The existence of antichiral surface band is related to the Zak phase of the bulk crystal. In Supplementary Note 3, we calculate Zak phases accumulated in the k_y direction, since the crystal is truncated in the y direction. In the surface BZ (Fig. S7c), the Zak phase is π inside two projected NLs while it is 0 outside, indicating a protected surface band between two NLs. The sudden jump of the Zak phase is due to the π Berry phase carried by DNLs.

Interestingly, the topological feature of the DNL is also manifested in the reflection phase winding near DPs [Deng, et al., Light: Sci. Appl. 11:134 (2022)]. This property is numerically verified in our metacrystal (Supplementary Note 5 and Fig. S9, see below). Since the existence of surface state should satisfy the relation of $\phi_{PC} + \phi_{PEC} = 2N\pi$, this 2π -winding can ensure at least one surface band emerged from the DP, no matter the reflection phase of the other crystal.

Fig. S7 **a** Bulk band structure on the plane of $k_z = 0.2\pi/d$. **b** Distribution of Zak phase sum ($\theta_1 + \theta_2$) accumulated in k_y direction, when $k_z = 0.2\pi/d$. **c** Zak phase distribution in the whole surface Brillouin zone.

Fig. S9 Reflection phase near Dirac point. Top view (a) and side view (b) of the setting to calculate the reflection phase. c Projected bulk band near the Dirac point with k_z equals zero. d-e Reflection phase at $k_x = 0.6\pi/a$ and $k_x = 0.7\pi/a$.

Comment 2: It is a bit difficult to understand from which is the plane of the investigations in Fig. 3 and 4. The Authors refer to the center of the bulk crystals. Is it the same as the center of the cell? An image of the cross-section would help a lot.

Response:

We thank the reviewer for this suggestion. In Fig. 3, we have added a purple line (Fig. 3a) and a purple rectangle (Fig. 3b) to clearly show the measured plane where we measured the bulk states. We measured the electric field on this plane by plugging the probe antennas into the reserved air holes (purple circles in the inset of Fig. 3a). On the other hand, in Fig. 4, we have highlighted the upper and the lower surfaces with blue and green lines, respectively. Detailed experimental setups have been added in Fig. S13 and Supplementary Note 7. The measured plane for the upper surface is depicted in Fig. S13e.

Fig. 3 Experimental observation of bulk ANLs. **a** Photograph of the sample. The purple line depicts the measured plane to probe the bulk states. Inset: enlarged top view of the structure. Purple circles depict the reserved air holes for probing. **b** Experimental setup for probing the bulk band structure. A dipole source antenna (yellow) is inserted from the bottom, whereas the probe antenna (green) scans the middle x - z plane hole-by-hole. A numerically simulated E_z distribution at 9.5 GHz is plotted at the measured plane. **c** Bulk Brillouin zone and its surface projection (yellow plane). The ANLs at $-HH$ and $-H'H'$ are projected to $-\tilde{H}\tilde{H}$ and $-\tilde{H}'\tilde{H}'$ of the surface BZ, respectively. **d-h** Simulated projected bulk bands on different k_z planes. The blue and red dots denote the Dirac points at \bar{K} and \bar{K}' points, respectively. Because the system is time-reversal-invariant, the frequency shifting condition is reversed for opposite k_z . **i-m** Measured projected bulk bands. The green dot-dashed lines outline the simulated band edges.

Fig. 4 Experimental observation of twisted ribbon surface state. **a** Experimental setup for probing the surface dispersion. The upper/lower surface is covered with an aluminum plate (deemed a PEC in the microwave regime). The probe antenna scans the crystal surface hole-by-hole. **b** Bulk Brillouin zone and its surface projection (yellow plane). **c** Simulated surface dispersion in the shape of a twisted ribbon. For clarity, the projected bulk bands are concealed, leaving only the surface bands. **d-h** Simulated projected band structure on different k_z planes. The antichiral surface bands for the upper and lower surfaces are highlighted by the cyan and green dashed lines, respectively. The purple line denotes a trivial surface band that is not protected by the nontrivial Berry phase of Dirac point. **i-r** Measured surface dispersion. The white lines represent the simulated bulk bands.

Fig. S13 Detailed experiment setup. **a** Top view of the sample. **b** The zigzag surface that supports the antichiral surface state. **c** Side view of the sample. **d** Experimental setup for probing the antichiral surface state. A metallic plate is placed beside the sample to serve as a PEC. The probe antenna scans the electric field profile at each outmost reserved air hole (highlighted in Fig. S13e). **e** An enlarged view of the sample, showing the detailed metallic bodies in unit cell, the reserved air holes for plugging antennas and the surface truncation.

Comment 3: How many points do the measurements along the x - z plan include? Could the authors provide the field data before the FFT, indicatively, for some frequencies?

Response:

When measuring the electric fields on the x - z plane, there are 51 measuring points in the x direction (with interval of 10 mm) and 81 points in the z direction (with interval of 3.5 mm). To demonstrate the antichiral surface transport clearly, we illustrate how we process the electric field data detailly in Fig. S14. As shown in Fig. S14a, we first measure the surface electric field in experiment. Then, we perform a Fourier transform to extract modes with different k_z components and integrate the rightward propagating energy and the leftward propagating energy, respectively (dashed rectangles in Fig. S14b show the region where we integrate the electric field).

In addition, a few indicative measured field profiles are provided in Figs. S15 of Supplementary Information. Although the electric fields on the right are obviously stronger than the ones on the left (indicating a unidirectional transport on the surface), the rightward propagating surface waves gradually decay in the x direction. The reason may be the imperfection of our sample. Since the sample is constructed by stacking PCBs in the z direction. The PCBs' out-of-flatness would lead to air gaps between neighboring boards and deteriorate the periodicity in the z direction. Therefore, the excited surface waves with certain k_z could be scattered to bulk states with other k_z , resulting in the attenuation of the antichiral surface waves.

Fig. S14 Data possessing for calculating the directionality of surface transport.

Fig. S15 Antichiral surface transport with different k_z component at different surfaces.

Comment 4: There are some significant qualitative differences in the simulated and experimental data for Figure 3. For example in Fig.3j there seems to be an additional band around 3GHz which is not predicted in theory. Are Fig.3i and Fig.3m supposed to

be mirrored? Why is it so different in experiment? In Figure 4k, the symmetric case, there is an asymmetry in the high frequencies. Could the Authors comment on that?

Response:

(1) There are some differences between the simulation and experimental data at around 3.5 GHz in Fig. 3. We find that these additional signals come from the transmission line modes between the source coaxial cable and probe coaxial cable. To see this, Figure R2a below plots the measured field at 3.5 GHz, where we can see that most of the signals are localized near the cable connecting source antenna. Corresponding FFT spectra are shown in Fig. R2c. Then, we perform a Fourier transform on the electric field around the source antenna (Fig. R2b). The resulting Fourier spectrum (Fig. R2d) corresponds to the modes at 3.5 GHz in Figs. 3i & 3j.

(2) Although the projected bulk bands for opposite k_z components (Figs. 3i & 3m) are supposed to be mirrored, their experimental FFT results may not be mirrored due to their different excitation efficiencies (e.g., usually determined by the source position, mirror symmetry of the finite size sample, sample imperfection and other experimental setup). Affected by the transmission line modes between the source and probe coaxial cables, the excitation efficiencies are different for positive and negative k_z components. On the other hand, the excitation efficiencies for positive and negative k_x components are different because our metacrystal does not have mirror symmetry in the x direction. Therefore, even for the symmetric case with zero k_z (Fig. 3k), there is some asymmetry in the experimental FFT results.

Fig. R2 **a** Measured E_z field profile at 3.5GHz before FFT. **b** E_z field profile near the source antenna. Only the field in the region highlighted in dashed rectangle is reserved while field in the other region is set to be zero. **c** Fourier spectrum corresponding to original field data with $k_z = -0.2\pi/d$. **d** Fourier spectrum corresponding to the transmission line modes with $k_z = -0.2\pi/d$.

Comment 5: The authors mention that “we consider the zigzag surface of the 3D hexagonal metacrystal by placing a metallic plate next to the sample”. This needs to be further clarified and discussed in the manuscript. What does this mirror symmetry imposed in the electromagnetic fields? Which is the termination of the photonic crystal before the PEC wall selected for the case appearing in figure 4?

Response:

We thank the reviewer for this helpful suggestion. To clarify the crystal surface we consider, we have revised Fig. 4 to illustrate the experimental setup for surface measurement. Besides, detailed experimental setup including the surface truncation of the metacrystal have been added in Fig. S13. Relevant information has also been added in the manuscript accordingly.

The aluminum plate used in our experiment can be viewed as perfect metal approximately in microwave regime. It requires the parallel electric fields to vanish on the PEC wall.

Fig. S13 Detailed experiment setup. **a** Top view of the sample. **b** The zigzag surface that supports the antichiral surface state. **c** Side view of the sample. **d** Experimental setup for probing the antichiral surface state. A metallic plate is placed beside the sample to serve as a PEC. The probe antenna scans the electric field profile at each outmost reserved air hole (highlighted in Fig. S13e). **e** A enlarged view of the sample, showing the detailed metallic bodies in unit cell, the reserved air holes for plugging antennas and the surface truncation.

Comment 6: Is the probe antenna in Figure 4 placed in between the PC wall and the sample?

Response:

In the measurement of surface state, the probe antenna is plugged into a series of air holes near the crystal surface (dashed rectangle in Fig. S13e). To make this clear, we

have added an experimental setup for surface probing as the new Fig. 4a. Detailed descriptions about experimental setups have been added in Supplementary Note 7.

Comment 7: Is the BZ presented in Fig3c as it is mentioned in Line 162 or in Fig4c?

Response:

Yes, bulk state measurement (Fig. 3) and surface state measurement (Fig. 4) share the same BZ and surface BZ. To clarify this, we replot the surface BZ in the new Fig. 4b, but with a different viewing angle.

Comment 8: Regarding the surface transport, it hard to understand where the surface mode propagates in the simulation. Is a 3D system considered in the simulation? Is the cross-section of the field in Fig.5 assumed just above the structure?

Response:

In the simulations of Fig. 5, we simulate a finite 3D metacryystal with the size of $16 \times 5 \times 10$ periods. The cross-section views in the original Figs. 5c-f are cut on an internal x-y plane inside the crystal, in order to show the localization of surface waves. To avoid misunderstanding, we replot these profiles with an oblique viewing angle in the new Figs. 5b and 5h. Besides, we have added the simulated field profiles on the upper and the lower surfaces (x-z planes) in the new Figs. 5c, 5d, 5i and 5j.

Fig. 5 k_z -dependent antichiral surface transport. Sketch of the energy flow for bulk modes (cyan arrows) and surface modes (orange arrows) with positive k_z (a) and negative k_z (g). **b** Simulated intensity profiles for $k_z = 0.2\pi/d$ at 9.5 GHz. The cyan cylinders represent the line sources. On both surfaces, the electromagnetic waves with positive k_z copropagate in the $+x$ direction. **c-d** Simulated E_z profile on both surfaces for $k_z = 0.2\pi/d$. **e-f** Directionality spectra in the frequency regime of the antichiral

surface state for $k_z = 0.2\pi/d$. The frequency regime corresponding to the antichiral surface state is highlighted in yellow. **h** Simulated intensity profiles for $k_z = -0.2\pi/d$. On both surfaces, the electromagnetic waves with negative k_z copropagate in the $-x$ direction. **i-j** Simulated E_z electric field profile on both surfaces for $k_z = -0.2\pi/d$. **k-l** Directionality spectra for $k_z = -0.2\pi/d$.

Comment 9: It would be interesting if the authors showed the scanned electric field data before the FFT in Figure 5.

Response:

We thank the reviewer for this suggestion. We have added the measured surface field before FFT in Fig. S14a, at 9.5GHz for example. We can see a strong signal localized near the source coaxial cable, which is due to the transmission line modes between the metallic plate and the metallic cable. Therefore, the surface waves were concealed by this strong signal. To show more details about the processed data, measured surface field profile for different k_z after FFT are added in Fig. S15.

Fig. S14 Data processing for calculating the directionality of surface transport.

Comment 10: It would be better if the term time reversal invariant would be appearing in the title.

Response:

We thank the reviewer for this helpful suggestion. We have revised the title to be “Antichiral surface states in time-reversal-invariant photonic semimetals”.

Comment 11: The quality of the figures needs to be improved.

Response:

We thank the reviewer for this suggestion. To illustrate our idea clearer, we have carefully improved the qualities of all the figures in manuscript.

Comment 12: A careful proof reading is necessary.

Response:

We thank the reviewer for this suggestion. We have carefully proofread and revised our manuscript.

REVIEWERS' COMMENTS

Reviewer #1 (Remarks to the Author):

All of my comments have been addressed.

Reviewer #2 (Remarks to the Author):

In the revised paper, the authors have made significant improvement upon the manuscript, satisfactorily matching the comments and suggestions by the reviewers. Thus I think the current version of manuscript can be accepted for publication in NC.

Reviewer #3 (Remarks to the Author):

The authors successfully address all my comments. I have no more questions and recommend its publication on Nature Communications.

Reviewer #4 (Remarks to the Author):

The Authors took into account the comments of the Reviewers and improved the manuscript in a good manner particularly for the theoretical part. They also provided convincing answers regarding the experimental part. As a last point, for consistency, I would suggest that the Authors added a comment from their response in the main manuscript regarding some differences between the simulation and experimental data of Fig.3.

Response to Reviewer #4

Comment: The Authors took into account the comments of the Reviewers and improved the manuscript in a good manner particularly for the theoretical part. They also provided convincing answers regarding the experimental part. As a last point, for consistency, I would suggest that the Authors added a comment from their response in the main manuscript regarding some differences between the simulation and experimental data of Fig.3.

Response: We thank the reviewer for his/her affirmation of our revised manuscript and the helpful suggestion. The discussion about the differences between simulations and experimental results in Fig. 3 has been added in the revised manuscript (Lines 168-171).